# Preparation of Quaternary Amphiphilic Block Copolymer PMA-*b*-P (NVP/MAH/St) and Its Application in Surface Modification of Aluminum Nitride Powders

**DOI:** 10.3390/molecules26195884

**Published:** 2021-09-28

**Authors:** Yu Wang, Guangdong Zhu, Shun Wang, Jianjun Xie, Zhan Chen, Ying Shi

**Affiliations:** 1Department of Electronics and Information Materials, School of Materials Science and Engineering, Shanghai University, Shanghai 200444, China; Wangliuliu2020@163.com (S.W.); xiejianjun@shu.edu.cn (J.X.); 2Shanghai Yuking Water Soluble Material Tech Co. Ltd., Shanghai 201318, China; sevenz@unipolymer.com (G.Z.); chenzhn621@unipolymer.com (Z.C.)

**Keywords:** RAFT polymerization, *N*-vinyl pyrrolidone, amphiphilic block copolymer, AlN powders, surface modification, hydrolysis resistance

## Abstract

Poly(methyl acrylate)-*b*-poly(*N*-vinyl pyrrolidone/maleic anhydride/styrene) (PMA-*b*-P (NVP/MAH/St)) quaternary amphiphilic block copolymer prepared by reversible addition-fragmentation chain transfer (RAFT) was used to improve the anti-hydrolysis and dispersion properties of aluminum nitride (AIN) powders that were modified by copolymers. Its structure was characterized by Fourier transform infrared spectroscopy (FT-IR) and Hydrogen nuclear magnetic spectroscopy (^1^H-NMR). The results demonstrate that the molecular weight distribution of the quaternary amphiphilic block copolymers is 1.35–1.60, which is characteristic of controlled molecular weight and narrow molecular weight distribution. Through charge transfer complexes, NVP/MAH/St produces a regular alternating arrangement structure. After being treated with micro-crosslinking, AlN powder modified by copolymer PMA-*b*-P(NVP/MAH/St) exhibits outstanding resistance to hydrolysis and can be stabilized in hot water at 50 °C for more than 14 h, and the agglomeration of powder particles was improved remarkably.

## 1. Introduction

Aluminum nitride (AlN) ceramics have a wide range of excellent properties, including high thermal conductivity, low dielectric constant and dielectric loss, and a thermal expansion coefficient that is similar to silicone, and are therefore considered a preferred material for hot sink devices such as the new-generation high performance ceramic wafer and electronic packaging [1,2,3]. AIN powder is the fundamental material used in the sintering process to create AIN ceramics. However, AIN powder can easily react with airborne water vapor, generating an AIN hydrolysate layer, and an increase in oxygen concentration can reduce the thermal conductivity of aluminum nitride ceramics [4]. Its hydrolytic feature also significantly inhibits the development of AIN ceramics’ water-based tape casting technique, thus the anti-hydrolysis property of AIN powders has become one of the industry’s prime difficulties.

The anti-hydrolysis treatment of AlN is generally to coat or produce a thin reaction layer of AlN particles via chemical bond or physical adsorption in order to prevent the AlN powder from hydrolyzing when exposed to water. The hydrolysis behavior of AIN powder in water can be effectively inhibited using commonly used methods such as coupling agent modification, phosphate acid, and other small molecule modifications [5,6,7,8]. However, problems remain in the wake of introducing silicone, phosphorus and other inorganic doped elements, and the powder after modification is not easy to water-based disperse. Low particle size and a highly dispersed slurry, on the other hand, are the foundations for the preparation of high-performance ceramics [6], and conventional anti-hydrolysis technologies struggle to match the powders’ anti-hydrolysis and high dispersion requirements.

Although water-soluble polymer has been used in the dispersion of ceramic powders to some extent, reports of AlN powder modification are uncommon. As polyvinyl pyrrolidone (PVP) contains high polar lactam groups and has strong binding force with hydroxyl, amino and carboxyl groups, plus not having selectivity in adsorption on ceramic powder surface, it will not change the particle crystal shape while reducing particle sizes, so it has a wide range of applications in powder dispersion [9].

Due to the strong reactivity of *N*-vinyl pyrrolidone (NVP) monomer, it is extremely easy to co-polymerize with other monomers containing vinyl unsaturated structure to generate polymer compounds with both NVP unit and other copolymer monomer structures [10,11,12,13], resulting in a product with a wide range of functional characteristics.

This paper is based on a mechanism of AIN powder surface modification for the design and synthesis of a quaternary amphiphilic block copolymer PMA-*b*-P(NVP/MAH/St) with alternating structures of NVP, Maleic anhydride (MAH), Styrene (St), Methyl acrylate (MA) using PVP as the main chain and RAFT polymerization. Both passivation and adsorption coating of the powder surface modification are achieved by chemical bonding between the lactam of NVP and the anhydride group of MAH in the molecular chain segment and the surface hydroxyl group of AIN powder. As a hydrophobic group, St, MA can form an amphiphilic block structure, improving powder hydrophobic properties and increasing hydrolysis resistance. At the same time, the steric hindrance of the two is combined with lactam and acid anhydride hydrolysis carboxyl electronegative charge repulsion, forming double particle protection. As a result, issues such easy powder aggregation and incomplete surface modification coating treatment are resolved.

## 2. Results and Discussion

### 2.1. Copolymer Design and Structure Analysis

In the design of copolymer, based on the modification mechanism of AlN powder, the stable adsorption of copolymer on the powder surface is enhanced on the basis of PVP lactam hydrogen bond adsorption by introducing maleic anhydride group on the basis of PVP structure. Acid anhydride has a good chemical bond with AlN powder surface -OH and -NH. Maleic anhydride monomer self-polymerization is challenging due to the higher steric hindrance, however, it is an electron acceptor and can form a charge transfer complex (CTC) [14,15,16,17] with the strong electron doner NVP and St as shown in Figure 1. Under the influence of the initiator, a copolymer with an excellent alternating structure is generated. Furthering the introduction of MA and St can effectively adjust the hydrophilic and hydrophobic features of the polymer, and on the basis of single pyrrolidone lactam electronegativity rejection dispersion, the abilities of stable adsorption and dispersion of copolymer to powder can be improved and the hydrolysis resistance of copolymer modified powder can be strengthened via methyl acrylate solvation chain, styrene steric hindrance structure, acid anhydride hydrolysis ionization repulsion.

One of the most important active polymerization technologies is RAFT polymerization. RAFT has rapidly gained popularity because of its benefits, which include a broad application range, gentle operating conditions, strong molecular structure designability, and a narrow product molecular weight distribution that is equivalent to classical free radical polymerization [18]. The molecular weight and molecular weight distribution of block polymers, as determined by GPC, are significant parameters for characterizing polymer properties and applications, as well as for investigating the regulating effect of reactive radical polymerization. The experiment analyzed and investigated the structure of the products via GPC and product element composition.

The experiment changed the ratio of NVP-MAH (CTC1) and St-MAH (CTC2) in the hydrophilic part of the (NVP/MAH/St) ternary component to study the copolymer structure. BCSPA was employed as the RAFT reagent to directly perform NVP/MAH/St ternary component RAFT polymerization in order to expedite the analysis of the copolymer component. Table 1 shows the copolymer composition.

As can be seen from the table above, the proportion of CTC1 and CTC2 has a significant impact on product composition. BCSPA was utilized to conduct RAFT polymerization with CTC1 in the absence of CTC2, and the reaction ceased after the initial commencement, demonstrating that BCSPA has no ability to control CTC1 produced by NVP-MAH. The fundamental cause of this behavior is because NVP is a non-conjugated monomer, and BCPSA trithioester is not an impact RAFT reagent [19,20] that does not have a polymerization effect, but does have a polymerization inhibitory effect.

With the addition of CTC2, the product now complies with the basic molecular weight and molecular weight distribution features of RAFT polymerization. At the same time, when CTC1: CTC2 > 1, the reaction conversion rate terminated after the consumption of CTC2 and basically remained at CTC1:CTC2 = 1:1 reaction conversion rate. The acid value of the copolymer is around 530 mg/g, and the molar ratio of MAH in copolymers is close to 50%, indicating that MAH in copolymer demonstrates an obvious trend of alternating polymerization with NVP and St in the form of charge transfer complexes [21]. Combined with acid value and N content, it can be concluded that CTC1 and CTC2 are within a specific range of proportion, that during chain transfer agent of BCSPA, CTC2 and CTC1 show a trend of alternating polymerization, and the product structure has a certain regularity of alternating structure in the charge transfer complex. Meanwhile, for group 4, the sample was tested under different conversion rates, and the copolymer composition of the product can be seen from Table 2. The N content and acid value of the product were both very close to the theoretical value of the alternating structure, confirming the regular alternating structure of the product.

The above condition also exists in the (NVP/MAH/St) chain extension experiment employing PMA-CTA, and PMA-CTA has no triggering impact on NVP/MAH (CTC1). The chain expansion reaction can only happen when CTC2 is introduced.

Table 3 illustrates the block reaction ratios of macromolecular chain transfer agent PMA-CTA (**a**) and (NVP/MAH/St) (**b**) in various proportions, as well as three amphiphilic block copolymers with varied molecular weights, where NVP:MAH:St = 1:2:1. The molecular weight distribution of the macromolecular transfer chain PMA-CTA is very narrow (PDI(*M*_w_/*M*_n_)1.17), showing that the produced chain transfer agent BCSPA has a good activity and regulated polymerization impact on MA, as shown in Table 3. The hydrophilic (NVP/MAH/St) part was introduced to carry out varying proportions of block copolymerization in the further chain extension, and the PDI value of the products obtained was 1.3–1.6. The ternary components’ activity was regulated by the macromolecular chain transfer agent, and the molecular weight and molecular weight distribution were within the controllable range. However, the (NVP/MAH/St) ternary system in the RAFT polymerization process cannot fully follow the alternating structure of charge transfer complex, resulting in a wider molecular weight dispersion problem after chain extension. There is also a minor amount of free radical polymerization, which can result in a wider molecular weight dispersion if the two polymerization processes are combined.

The experiment comprised sampling inspection over the A3 process at various conversion rates, with PMA-CTA excluded from the total amount and N content calculated. As shown in Figure 2, N content essentially maintained a balance at various conversion rates. The polymerization process indicated a pattern of alternate polymerization of CTC1 and CTC2, which was compatible with the hypothesized alternate polymerization. When the conversion rate exceeds 70%, the nitrogen content decreases slightly. The fundamental reason for this is because when the conversion rate rises, the concentration of CTC1 drops and the system’s viscosity rises, making it difficult to maintain the alternating copolymerization trend. There is a certain random polymerization tendency in addition to the alternating copolymer. However, the overall structure remains relatively regular and adheres to the alternating composition. As a result, the planned and produced amphiphilic block copolymers exhibit an alternating distinctive structure, PMA-*b*-[(NVP-MAH)-alt-(St-MAH)].

### 2.2. FT-IR Characterization

Figure 3 shows the infrared spectrum of PMA-*b*-P(NVP/MAH/St) amphiphilic block copolymer, which indicates that the absorption peaks of 3062 cm^−1^ and 3030 cm^−1^ are the CH stretching vibration absorption of the benzene ring, and 756 cm^−1^ and 705 cm^−1^ are the out-of-plane bending vibration absorption of the benzene ring, proving the existence of styrene structure. The absorption peaks of 1850 cm^−1^ and 1780 cm^−1^ are the symmetric and asymmetric stretching vibration absorption of C=O in maleic anhydride. The absorption peaks of 1224 cm^−1^ are the stretching vibration absorption of C-O in five-member cyclic anhydride, which proves the existence of maleic anhydride in polymer. The absorption peak 1733 cm^−1^ is the stretching vibration absorption of C=O in methyl acrylate, which proves that the polymer contains methyl acrylate. The stretching vibration absorption peak of C=O in the PVP lactam group is 1679 cm^−1^, and the stretching vibration absorption peak of C-N in the lactam group is 1288 cm^−1^, indicating that the polymer has NVP structure. The amphiphilic block copolymer PMA-*b*-P (NVP/MAH/St) has been effectively manufactured, as evidenced by the product’s purification procedure and the typical absorption peaks of the aforesaid polymers.

### 2.3. H-NMR Characterization

Figure 4 shows ^1^H-NMR (δ, ppm, DMSO) of PMA-*b*-P (NVP/MAH/St) block copolymer, in which δ = 0 is the proton peak of the internal standard tetramethylsilane (TMS) and δ = 2.50 is the solvent peak of the solvent DMSO. 6.81–7.60 belongs to the characteristic chemical shift of hydrogen (**a**) of benzene ring in St structure, 3.85–4.2 belongs to the characteristic chemical shift of hydrogen (**c**) on methine in NVP backbone chain, and 3.65 belongs to the characteristic chemical shift of methyl hydrogen (**b**) connected with oxygen in MA. Additionally, 3.31 characteristic chemical shifts of hydrogen (**d**) on the methine of the MAH backbone chain and hydrogen (**e**) on the methylene associated with N on the pyrrolidone heterocyclic ring; 1.49–2.46 are the characteristic chemical shifts of hydrogen (**h, g**) on the methine of the main chain of MA and St, and hydrogen (**f**) on the methylene linked to the carbonyl group on the pyrrolidone hybridization, respectively. Then, 0.74–1.31, belongs to the characteristic chemical shifts of methylene hydrogen (**k, j,**
**i**) on the main chains of MA, St and NVP, and hydrogen (**r**) on the ring of NVP pyrrolidone, respectively. The spectra reflected the corresponding hydrogen spectral chemical shifts in the composition of the polymerization products. It was confirmed that the final synthesis product was the block copolymerization product of PMA-*b*-P (NVP/MAH/St) when the FT-IR analytical findings were combined.

### 2.4. AlN Powder Modified by PMA-b-P (NVP/MAH/St) Block Copolymer

#### 2.4.1. Hydrolysis Test

In water, AIN powder is easily degraded and hydrolyzed into Al(OH)_3_ and NH_3_, causing pH to rise. As a result, one of the important indicators for measuring the anti-hydrolysis of the powder is the investigation of pH change in the aqueous phase. According to the modification handling method, A1 block copolymer was used to modify the surface coating of AlN powder, then 2 g modified AIN powder was added into 50 mL deionized water and stirred evenly, the powder was placed in an 50 °C baking oven for heat preservation and the pH of the suspension system was monitored over time. Figure 5 shows the change in pH of the suspension created in water by unmodified/modified/micro-crosslinked AIN powder at 50 °C over time, demonstrating that unmodified powder, in the water batch at 50 °C, the pH will approach more than 10 within 1 h, indicating that hydrolysis is nearly complete. The pH of the copolymer-modified powder remained constant at around 6.0. In the beginning, there was a drop in pH due to unreacted anhydride hydrolysis. The pH has a noticeable tendency to rise after 8 h. Water molecules diffused through the coating layer generated in the copolymer powder surface, permeated into the surface of AIN powder, and hydrolysis happened as a result of Brownian movement. The micro-crosslinking treatment was achieved after micro-crosslinking with hexylenediamine, which inhibited the movement of water molecules and increased the coated layer’s durability. The pH maintained basic stability within 14 h, showing good hydrolysis resistance.

#### 2.4.2. XRD Analysis

The phase study of AIN powder before and after hydrolysis can be done using the XRD spectrum. The XRD patterns of AIN powder modified by block copolymer at various times are shown in Figure 6. Figure 6A(a) shows the diffraction patterns of original AIN, and Figure 6A(b) shows the diffraction patterns of AlN powder modified by block copolymer after 8 h hydrolysis, and it can be seen that pattern a and b are completely the same without any new diffraction peak, indicating that the modification of AlN powder by block copolymer has no effect on the phase lattice of AIN, and the modification only occurs on the surface of the powder. Additionally, AIN powders modified by block copolymer showed a certain resistance to hydrolysis at 50 °C within 8 h. Figure 6A(c) is the diffraction pattern of AlN powder after hydrolyzing for 14 h, and its phase is obviously changed. It has been hydrolyzed to Al(OH)_3_ and AlO(OH). Water molecules infiltrate and diffuse into AlN powder, destroying the covering created via copolymer modification. Figure 6B shows the XRD pattern of AlN powder following micro-crosslinking treatment at various hydrolysis time. Figure 6B(a) is the original diffraction pattern, and Figure 6B(b) and Figure 6B(c) are the diffraction pattern of AlN powder modified by micro-crosslinking after 7 h and 14 h hydrolysis, from which it can be observed that, after 7 h and 14 h hydrolysis, AlN powder diffraction pattern is basically consistent with the original AlN powder. Meanwhile, after hydrolyzing at 50 °C for 14 h, only a trace of AlO(OH) appeared, and there’s no impact on the phase lattice of AlN powder after micro-crosslinking treatment, and no other new diffraction peak occurred, which indicated that the surface of AlN powder after undergoing micro-crosslinking treatment was more tightly coated, and the diffusion movement of water molecules was limited by crosslinking treatment, which is more conducive to the improvement of its hydrolysis resistance.

#### 2.4.3. SEM, TEM and EDS Analysis

Figure 7 shows the SEM and TEM topography of AlN powder. Among them, (a) shows the SEM of the original AlN powder, which is in the shape of an irregular smooth sphere with relatively close agglomeration between particles. (b) shows SEM after hydrolysis of AlN powder, demonstrating wrinkle and irregular character, as well as considerable morphological changes. (c) shows the SEM of AlN powder modified by block copolymer, which is essentially the same as the original AlN, and the surface is relatively smooth and irregular, demonstrating that the modification of block copolymer will not change the phase of AlN powder, and the particles agglomeration has been improved to some extent. (d) shows the SEM of the modified AlN powder that has been micro-crosslinked. The morphology is the same as that of the original AlN powder, but the agglomeration between particles is significantly improved, indicating that the micro-crosslinked treatment improves particle repulsion and steric hindrance, and particle surface modification is more complete. This is also one of the reasons why the micro-crosslinking treatment is better for improving AlN powder’s hydrolysis resistance. (e) shows the AlN/copolymer core-shell with a thin copolymer film within a uniform thickness of about 8 nm, the block copolymer was well bonded and coated on the surface of AlN powder.

The EDS element distribution of a block copolymer covered with modified AlN powder is shown in Figure 8. As modified AlN powder was coated on aluminum foil for EDS testing, the distribution of Al element was denser than that of N element, as seen in Figure 8c,d. Furthermore, the C and O components from the copolymer can be observed in Figure 8e,f, where they are evenly dispersed on the surface of the aluminum nitride particles without accumulation, demonstrating that the block copolymer uniformly encapsulates AlN powder.

## 3. Materials and Methods

### 3.1. Materials

NVP (Shanghai Yuking Water Soluble Material Tech Co. Ltd. (Shanghai, China), 99.5%) were purified by vacuum distillation; St (99%) were purified by vacuum distillation; methyl acrylate (98.5%) were purified by vacuum distillation; and MAH (99.5%), Azobisisobutyronitrile (AIBN), 1,4-Dioxane, and 0.5 μm AlN powder were obtained from Shanghai Macklin Biochemical Tech Co., Ltd., Shanghai, China. Other reagents or analytical grade solvents were sourced from commercial resources.

### 3.2. Characterization of the Copolymer

FT-IR: The structure of copolymer was characterized by FT-IR (KBr, Shimadzu Fourier transform infrared spectrometer IRSpirit-T (Shimadzu Corporation, Kyoto, Japan), and the range of scanning is between 4000–500 cm^−1^.

^1^H-NMR: The ^1^H-NMR of copolymer was characterized by Bruker Advance spectrometer (Bruker Corporation, Billerica, MA, USA) at 400 MHz, and CDCl_3_ or DMSO as solvent.

GPC: GPC was used to determine the molecular weight and polymer dispersion index of all polymers used in this work, and Shimadzu GPC (Shimadzu Corporation, Kyoto, Japan) was employed for analysis. The copolymer was neutralized and dissolved in an appropriate amount of sodium hydroxide solution in a water bath of 60–70 °C, with pH adjusted to 7–9, and filtered through a 0.45 μm filter membrane. The mobile phase employed 30% acetonitrile aqueous solution of 0.6%NaCl, flow rate of 1.0 mL/min, column temperature of 40 °C and monodisperse PEG/PEO as reference standard to generate the calibration curve.

### 3.3. Experimental Process

#### 3.3.1. Synthesis of Chain Transfer Agent 2-(((tert-butyl thio)carbonothioyl)thio) Propanoic Acid (BCSPA)

Following the method as described in the reference literature [22,23], tert-butyl mercaptan (18.0 g, 0.2 mol) and 30 mL distilled water were added to a round-bottomed flask, and 20.0 g sodium hydroxide solution with a mass fraction of 40% was added to the flask after proper stirring. Then, 10 mL acetone was added to get a clear colorless solution, stirred for 30 min and cooled to room temperature. Carbon disulfide (17.2 g, 0.23 mol) was added to get an orange solution. Stirring for 30 min in the ice bath, then 2-bromopropionic acid (31.4 g, 0.21 mol) and 20.0 g sodium hydroxide solution with 40% mass fraction were slowly added in order, the temperature was controlled below 30 °C, and stirred for 24 h at room temperature; at the end of the reaction, distilled water was added to dilute it. Under the circumstance of an ice bath, 30 mL concentrated hydrochloric acid was added to get a yellow oil layer, and stirred until the oil layer solidified and plenty of yellow particles were precipitated out. The solidified oil layer was filtered out, washed with a large amount of distilled water, dried for 24 h in a vacuum oven to get 40 g yellow powder, and then recrystallized with *n*-hexane for 3 times to obtain 35 g yellow solid. ^1^H-NMR (δ, ppm, CDCl_3_): 10.89 (br, 1H, COOH), 4.77–4.81 (q, 1H, -SCHCH_3_), 1.59–1.63 (q, 3H, C**H**_3_CS, SCHCH_3_).

#### 3.3.2. Synthesis of Macromolecular Chain Transfer Agent PMA-CTA

In nitrogen atmosphere, 10 mL of dioxane, MA 10.3 (120 mmol), BCSPA 1.43 g (6 mmol), and AIBN 0.2 g (1.2 mmol) were added into the flask. After bubbling in nitrogen for 30 min, the flask then reacted at 70 °C for 5 h. The reaction was terminated in an ice water bath, precipitated in ethanol/water (7:3), and a small amount of ethanol was re-dissolved. After repeated dissolution-precipitation operations for 2–3 times, a relatively pure product was obtained. The product was dried at 50 °C in a vacuum drying oven to obtain yellow viscous substance under normal temperature with a monomer conversion rate of 79.2%.^1^H-NMR (δ, ppm, CDCl_3_): 4.81 (-CH_2_CHS-), 3.66 (-OCH_3_), 2.22–2.64 (-CH_2_CH(COOCH_3_)-), 2.08–2.22 (-CH(CH_3_)-), 1.42–1.86 (-CH_2_CH(COOCH_3_)-), 1.15–1.26 (-C(C**H**_3_)_3_), 0.82–0.90 (-CH(COOH)CH_3_), *M*_n__,GPC_ = 1310 g·mol^−1^, *M*_w_/*M*_n_ = 1.17.

#### 3.3.3. Synthesis of Amphiphilic Block Copolymer PMA-*b*-P (NVP/MAH/St)

In nitrogen atmosphere, PMA-CTA 1.3 g (1 mmol), AIBN 0.033 g (0.02 mmol), and appropriate amounts of NVP, MAH, St and dioxane, were added into the flask. After being nitrogen bubbled for 30 min, the oil batch temperature rose up to 60 °C, and reacted for 12 h. After cooling in an ice water bath, the reaction was terminated, and the product was precipitated in ethanol. After the dissolution of DMSO, the pure product was obtained by repeated dissolution-precipitation operations for 2–3 times. Then, after being washed with hot toluene several times, centrifugal separation was performed. The yellow powder was obtained by drying at 50 °C in a vacuum drying oven. Figure 9 shows the synthesis process of block copolymer PMA-*b*-P (NVP/MAH/St).

### 3.4. Effect Verification of AIN Powder Surface Modification

#### 3.4.1. Modified Treatment

In 100 mL ethanol, 10 g AIN powder and 1 g block copolymer were mixed, dispersed by 2000 r/min high-speed shearing for 10 min, followed by ultrasonic treatment for 5 min, and then modified by stirring for 6 h in an oil batch at 80 °C. Removed and centrifuged at 4000 r/min for 3 min, then washed the dispersed precipitate with ethanol, centrifuged again, repeated treatment for 2–3 times, and vacuum dried at 50 °C. Micro-crosslinking treatment: After the powder reacted with block copolymer at 80 °C for 6 h, 0.2 g hexylenediamine was added, and heat preservation continued and stirred for another 2 h. Taking down, centrifuged at 4000 r/min for 3 min, washed the dispersed precipitate with ethanol, centrifuged again, repeated treatment for 2–3 times, and vacuum dried at 50 °C.

#### 3.4.2. Hydrolysis Test and Characterization

First, 2 g of modified AlN powder were taken into 50 mL water, dispersed by ultrasonic treatment for 2 min. After being evenly dispersed, a hydrolysis experiment was carried out in an oven at 50 °C. The pH of the dispersion solution was recorded at various time intervals, and the modified powder hydrolysis resistance was characterized via X-ray diffraction (XRD), scanning electron microscope (SEM), transmission electron microscopes (TEM) and energy dispersive spectrometer (EDS).

## 4. Conclusions

(1) Using 2-(((tert-butyl thio)carbonothioyl)thio)propanoic acid as chain transfer agent and AIBN as initiator, based on the mechanism of surface modified powder via copolymer, the quaternary amphiphilic block copolymer of PMA-*b*-P (NVP/MAH/St) was successfully prepared by RAFT polymerization, and characterized by FT-IR and ^1^H-NMR structure. The copolymer was found to be the target chemical, with a narrow molecular weight distribution, predictable polymerization characteristics, and a regular alternating structure in the hydrophilic chain section of the product.

(2) The surface of AlN powder was modified using the amphiphilic block copolymer that had been synthesized. It remained stable in water at 50 °C for more than 14 h after micro-crosslinking, demonstrating strong hydrolysis resistance. The modified powders were characterized by XRD, SEM, TEM and EDS. The surface modification treatment had no effect on the phase lattice of the AlN powders, and the copolymer was uniformly distributed on the surface of the powders, considerably improving the agglomeration phenomenon.

## Figures and Tables

**Figure 1 molecules-26-05884-f001:**
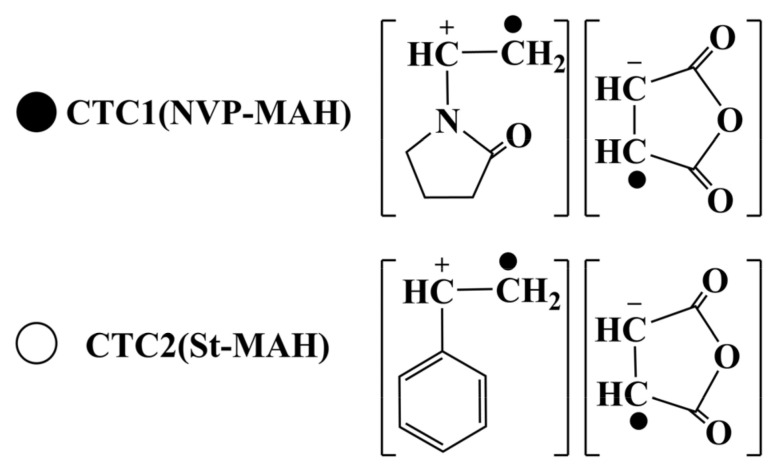
The CTC structure formed by MAH with NVP and St, respectively.

**Figure 2 molecules-26-05884-f002:**
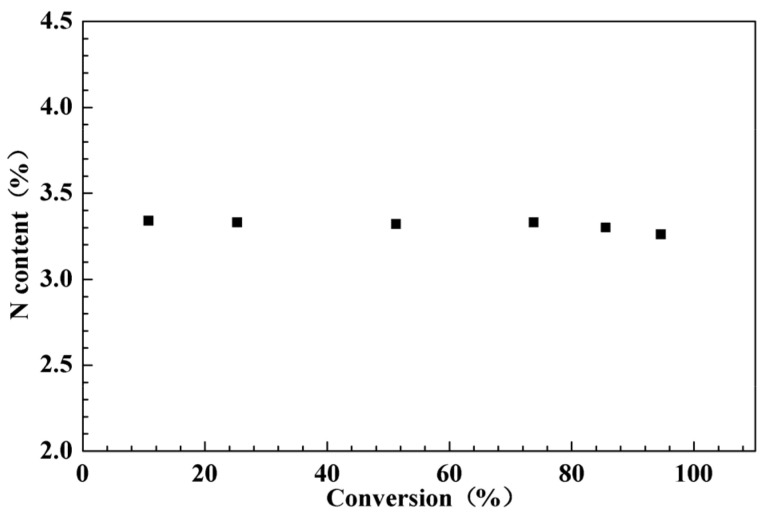
N content of the system at different conversion rates.

**Figure 3 molecules-26-05884-f003:**
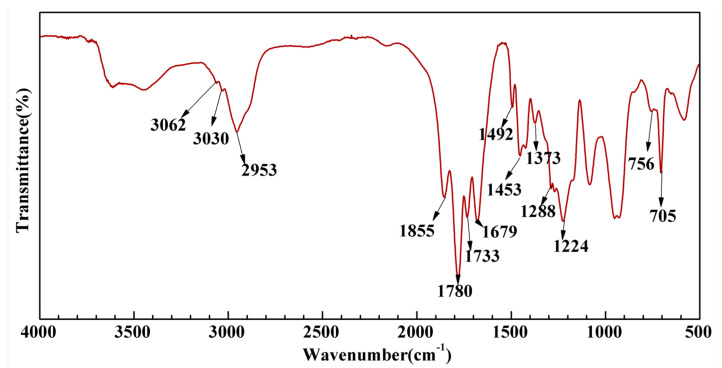
Infrared spectra of PMA-*b*-P (NVP/MAH/St) amphiphilic block copolymer.

**Figure 4 molecules-26-05884-f004:**
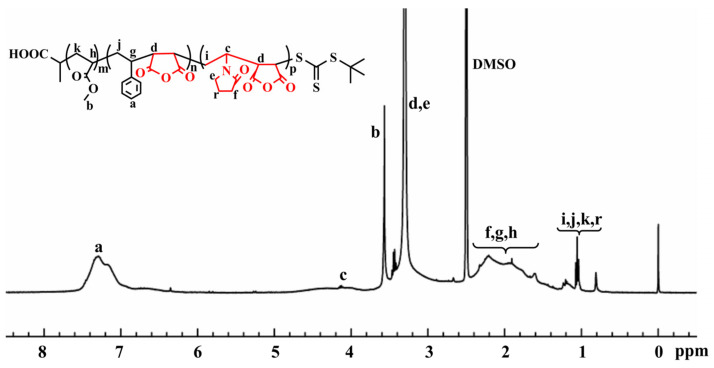
^1^H-NMR spectra of block polymer PMA-*b*-P (NVP/MAH/St).

**Figure 5 molecules-26-05884-f005:**
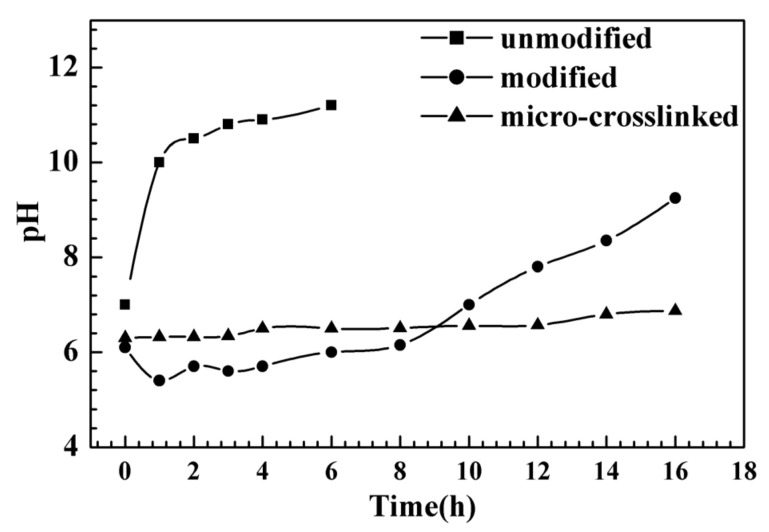
The pH change over time of original and surface modified AlN powder suspension at 50 °C.

**Figure 6 molecules-26-05884-f006:**
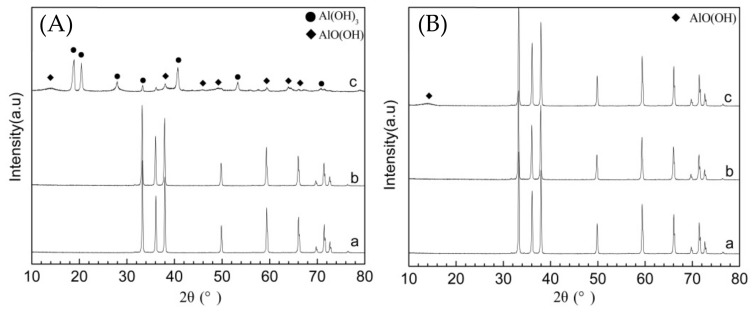
XRD patterns of AlN powders modified by block copolymer at 50 °C with different hydrolysis times: (**A**) a original AlN powders; (**A**) b hydrolyzed for 8 h modified powder; (**A**) c hydrolyzed for 14 h modified powder; (**B**) a original AlN powder; (**B**) b hydrolyzed for 7 h micro-crosslinking modified powder; (**B**) c hydrolyzed for 14 h micro-crosslinking modified powder.

**Figure 7 molecules-26-05884-f007:**
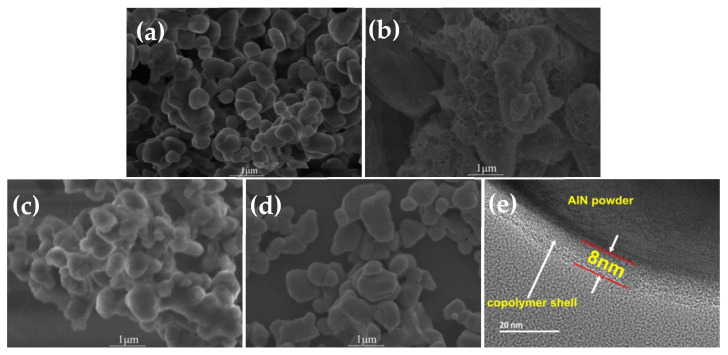
SEM topography of AlN powder: (**a**) Original AlN powder; (**b**) hydrolyzed AlN powder; (**c**) block copolymer modified AlN powder; (**d**) AlN powder modified by micro-crosslinking; (**e**) TEM topography of AlN powder modified by micro-crosslinking.

**Figure 8 molecules-26-05884-f008:**
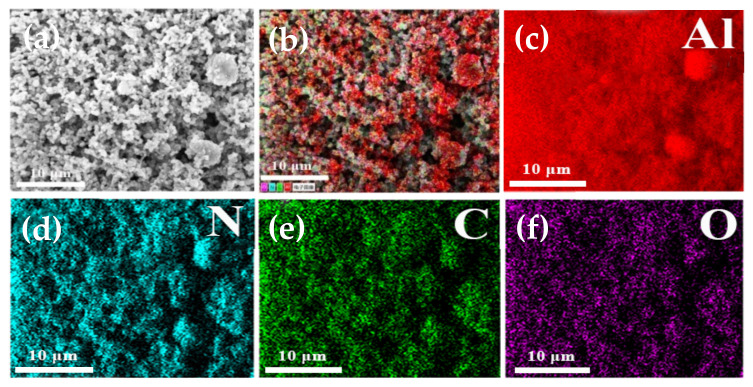
EDS diagram of block copolymer modified AlN powder: (**a**) Original EDS diagram; (**b**) EDS element hierarchy diagram; (**c**) Al element distribution; (**d**) N element distribution; (**e**) C element distribution; and (**f**) O element distribution.

**Figure 9 molecules-26-05884-f009:**
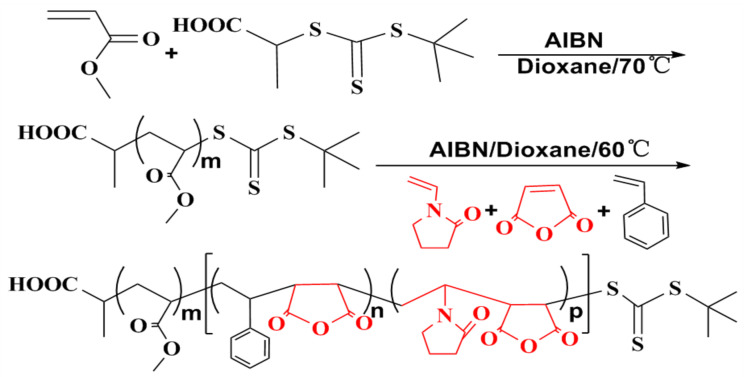
The synthesis process of PMA-*b*-P (NVP/MAH/St).

**Table 1 molecules-26-05884-t001:** Copolymer composition at various CTC ratios.

Group	Monomer Feed (mmol)			Conversion	Acid Value (mg/g)	N Content/%	Copolymer Composition (mol%)		*M*_n_ (*M*_w_/*M*_n_)
	BCSPA	CTC1	CTC2				CTC1	CTC2	
1	1	40	0	<5.6%	528	6.60	-	-	5300 (1.86)
2	1	30	10	49.6%	526	3.43	52.1	47.9	3730 (1.45)
3	1	24	16	78.3%	535	3.37	50.4	49.6	6150 (1.38)
4	1	20	20	95.6%	532	3.32	49.9	50.1	7715 (1.46)
5	1	16	24	98.3%	534	2.71	40.6	59.4	7648 (1.54)

**Table 2 molecules-26-05884-t002:** Copolymer composition in experimental group 4 at various conversion rates.

Conversion	9.6%	24.3%	51.3%	73.8%	95.6%
N content/%	2.68	2.98	3.15	3.35	3.31
Acid value (mg/g)	480	518	530	535	532

**Table 3 molecules-26-05884-t003:** The copolymer composition under various proportions between PMA and (NVP/MAH/St).

Copolymer	a (mmol)	b (mmol)	*M* _n_	M_w_/*M*_n_	N Content/%	Structure
A0	1	0	1310	1.17	-	MA_15_
A1	1	50	6020	1.49	2.62	MA_15_-*b*-(CTC1_12_-alt-CTC2_12_)
A2	1	80	8410	1.36	2.81	MA_15_-*b*-(CTC1_18_-alt-CTC2_18_)
A3	1	100	10500	1.57	2.93	MA_15_-*b*-(CTC1_23_-alt-CTC2_23_)

## Data Availability

Data has been indicated in the manuscript.

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
