# Peer review of "Preparation of Quaternary Amphiphilic Block Copolymer PMA-b-P (NVP/MAH/St) and Its Application in Surface Modification of Aluminum Nitride Powders"

_molecules, 2021, doi:10.3390/molecules26195884_

Round 1
Reviewer 1 Report
In the manuscript entitled “Preparation of quaternary amphiphilic block copolymer PMA-b-P (NVP/MAH/ St) and its application in surface modification of AlN powders”, a quaternary amphiphilic block copolymer was prepared by RAFT polymerization in order to improve the anti-hydrolysis and dispersion properties of AlN powders. The polymer synthetized was characterized by different techniques such as infrared spectroscopy, GPC, EDS, XRD and nuclear magnetic resonance spectroscopy. The GPC analysis indicates that the molecular weight distribution is controlled (1.35~1.60). The authors relates that NVP/MAH/St forms a regular alternating ar-rangement structure through charge transfer complexes. However, I am not sure that it is properly observed in the results. The work seems interesting. However, the text structure and English language is very problematic. I recommend that the text strucre and english language must be improved before publication.
imposrtant corrections and suggestions
1 – The english language must be hard revised. It is very important!
2 – Page 4, Figure 2. Image quality is not good and the chemical structures are not correct. The author must replace the figure, using a correct structure of the chemicals.
3 – Page 7, Figure 3 and 4. Image quality is not good. Replace the Figure 3 and 4.
4 – Page 8, Figure 5. The same fonts and size must be used in the letters and symbols.
5 – Page 8, Figure 5. The chemical shift around 6.2 ppm was not attributed by the authors. The author may explain the signals around 6.2 ppm.
6 – Page 9, Figure 6. Image quality is not good. Replace the Figure 6.
7 – The authors relates that NVP/MAH/St forms a regular alternating arrangement structure through charge transfer complexes. However, I am not sure that it is properly observed in the results. How the authors may prove the charge transfer complexes?
Minor corrections
1 – page 1, abstract section: The abbreviation names must not be used in the abstract section. The author must replace all abbreviation to the usual name of the compounds. Example: replace “St” to “styrene”; replace “AIN” to “aluminum nitride”
2 – Page 2: The section “2.Results”is not correct. Replcade the term “”2.results”to “2. Experimental section”.
3 – page 2, line 56: The name of the monomer was not used before in the text. replace “NVP monomer” to “N-vinyl pyrrolidone (NVP) monomer”.
4 – page 2, line 63: The monomer was mentioned before (after correction “1”). Its not necessary to write the monomer name. Replace “N-vinyl pyrrolidone(NVP) monomer” to “NVP monomer”.
5 – page 2, tine 64: the symbol PNVP was not mentioned before. The authors must specify the name of this compound.
6 – Page 2, results and instruments section: All text is not making any sense. The author must rewrite all text in this section. The compounds were distillated and purified? Was the purification of the compounds performed by distillation?
7 – page 2, line 91: Remove “and the stirring was uniform”. Its not necessary to inform readers about this things.
7 – page 2, experimental section: The verbs used in the text must be replace to the same period (past). Example: replace “add 10 ml …” to “10 ml was added …”
8 – page 3, line 104: The author related that nitrogen is used to protect the compounds… However its wrong. The nitrogen is used in the chemical media to avoid oxidation process when oxygen molecules is present in the chemical system.
Remove “With the protection of nitrogen” from the text. The author may write: The synthesis was performed in oxygen free, using nitrogen as inert media.
9 – page 4, lines 134-136: The author must rewrite all manuscript. The english language must be improved.
Observation: line 135: Its really funny! The word “Penetration” was used in the text. Is it serious??? Are we reading about chemistry science?
The paragraph found in 134-136 lines should be improved to “Gel Permeation Chromatography was used to determine the molecular weight and polymer dispersion index of all polymers used in this work.
Its important to say that GPC technique should be mentioned in the instruments section. Its not necessary to relate Shimatzu company or other specification every time.
Reviewer 2 Report
The are many problems with English and nomenclature, e.g.,
N-vinyl pyrrolidone should be N-vinylpyrrolidone.
monomer names should not be capitalized
The paper is very difficult to read and needs substantial rewriting.
There is literature on the maleic anhydride NVP copolymerization. See, for example, Polymer Bulletin volume 60, pages 741–752 (2008). This should be cited and the data compared.
Although there is good evidence that charge transfer complexes are formed the evidence that they are involved in polymerization is less clear. The literature on this should be consulted.
Round 2
Reviewer 1 Report
In the manuscript entitled “Preparation of quaternary amphiphilic block copolymer PMA-b-P (NVP/MAH/St) and its application in surface modification of AlN powders”, teh authors considered the suggestions and corrections to improve the quality of the text. After last corrections the manuscript is now ready to be published.
Author Response
Dear reviewers,
Thank you very much for your insightful suggestions on our manuscript titled“Preparation of quaternary amphiphilic block copolymer PMA-b-P (NVP/MAH/ St) and its application in surface modification of AlN powders”(Manuscript ID: molecules-1351347)”. Once again I am grateful to your professional guidance which has greatly helped us to finalize the paper.
We sincerely hope this manuscript will be finally accepted to be published in your famous journal.
Best regards
Dr. Yu Wang
Reviewer 2 Report
It would be better if the titles specified aluminum nitride powders rather than just the abbreviation.
line 17 delete " The molecular weight and molecular weight distribution of the polymer were determined by Gel permeation chromatography (GPC)." - it is not necessary to mention common techniques in the abstract.
The paper still requires substantial editing to improve the English.
line 34 - "a basic material" should be "the basic material"
For the RAFT agent used - name should be 2-(((tert-butylthio)carbonothioyl)thio)propanoic acid - the tert-butyl radical might also initiate polymerization. The structure of the polymer might not be as indicated.
